# Radar-Based Hand Gesture Recognition Using Spiking Neural Networks

**Ing Jyh Tsang [1,*], Federico Corradi [2] , Manolis Sifalakis [2] , Werner Van Leekwijck [1] and Steven Latré [1]**

1 IDLab-Department of Computer Science, University of Antwerp-IMEC, Sint-Pietersvliet 7,
2000 Antwerp, Belgium; werner.vanleekwijck@uantwerpen.be (W.V.L.); steven.latre@uantwerpen.be (S.L.)

2 IMEC Nederland, Holst Centre High Tech Campus 31, 5656 AE Eindhoven, The Netherlands;
federico.corradi@imec.nl (F.C.); manolis.sifalakis@imec.nl (M.S.)

* Correspondence: ingjyh.tsang@uantwerpen.be

**Abstract:** We propose a spiking neural network (SNN) approach for radar-based hand gesture recognition (HGR), using frequency modulated continuous wave (FMCW) millimeter-wave radar. After pre-processing the range-Doppler or micro-Doppler radar signal, we use a signal-to-spike conversion scheme that encodes radar Doppler maps into spike trains. The spike trains are fed into a spiking recurrent neural network, a liquid state machine (LSM). The readout spike signal from the SNN is then used as input for different classifiers for comparison, including logistic regression, random forest, and support vector machine (SVM). Using liquid state machines of less than 1000 neurons, we achieve better than state-of-the-art results on two publicly available reference datasets, reaching over 98% accuracy on 10-fold cross-validation for both data sets.

**Keywords:** radar; liquid state machine; spiking neural network; hand gesture recognition





## 1. Introduction

Hand gesture recognition has been an active field of research due to the quest to provide a better, more efficient, and intuitive mechanism for human–computer interaction (HCI). Many different sensor modalities have been used to address this challenge, such as radar [1–3], cameras [4,5], dynamic vision sensors (DVS) [6–9], or electromyography (EMG) systems [10,11]. Moreover, several different machine learning techniques have been proposed to address HGR, such as a convolutional neural network (CNN) [12], long short-term memory (LSTM) [12,13], and spiking neural networks [8,11].

While advancement in image recognition systems makes frame-based cameras propitious for HGR, it has disadvantages, such as the need for proper illumination conditions, camera position, and additional image processing tasks such as segmentation to isolate the gestures from the background scene. Alternatively, event-based cameras, i.e., DVS, are designed as a neuromorphic event-based vision system and a natural fit to be used with SNNs [6–9]. However, they remain a relatively costly device. Therefore, we have focused on a radar sensor as it is cost-effective, and its form factor has been miniaturized to the point of fitting in smartphones, wristbands, or headphone devices [12,14]. Moreover, the radar signal is robust against adverse weather or illumination conditions, and it offers a better privacy-preserving system than vision sensors [15].

This paper focuses on radar-based hand gesture recognition systems using spiking neural networks, and we use the Soli [12] and Dop-NET [2,16] as reference datasets, which both are based on FMCW millimeter-wave radar. Moreover, in particular, we used a liquid state machine, a type of reservoir computer capable of universal approximation [17,18], and we used different readout maps [19], or detectors [20] to compare and perform the classification tasks.

In summary, the contributions of this paper include:

1.  We introduce a novel radar signal to spike representation, which, when used as input to a liquid state machine with a classifier, achieves better than state-of-the-art results for a radar-based HGR system.
2.  We present a much simpler processing pipeline for gesture recognition than most state-of-the-art neural-network solutions, with a small footprint in terms of neurons and synapses, thus with great potential for energy efficiency. It is based on LSM, in which training is performed only at the readout layer, simplifying the overall learning procedure and allowing easy personalization.
3.  We conduct performance analysis of our approach, making extensive simulations to demonstrate how the size, input signal, and classifiers affect the system's overall capability.

## 2. Datasets, Spike Encodings, and Network Configuration

This section gives an overview of the radar-based hand gesture recognition pipeline. First, Section 2.1 provides some background in radar systems. In particular, we detail the FMCW pipeline and how range-Doppler or micro-Doppler data are obtained from this type of system. Moreover, Figure 1 depicts the process starting from the datasets, mapping the sections to the relevant steps of the pipeline. Sections 2.2 and 2.3 detail the used datasets. We describe the pre-processing and the neural encoding scheme to convert the radar signal to spike sequences required for the SNN in Section 2.4. Furthermore, Section 2.5 details the LSM architecture, the timing, and parameterization and explores two alternative ways the input spike representation can be encoded and used in conjunction with the LSM. Finally, it discusses possible optimizations of the system.

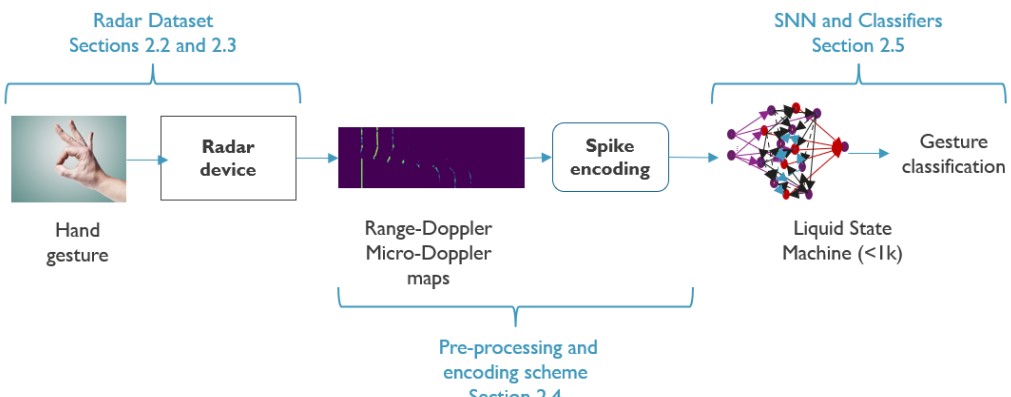

**Figure 1.** Pipeline of the radar-based hand gesture recognition system, from the dataset to the end classification, indicating the respective sections that detail each of the steps. The SNN and Classifier diagram is detailed in Section 2.5.

### 2.1. Radar System

Radar systems exploit electromagnetic waves to detect and locate objects in the environment. A minimal radar system consists of a transmitter, a receiver, and signal and data processing modules. While there are many different types of radar systems, Shahzad et al. [21] identifies two types generally used for HGR, i.e., Pulsed and Continuous-Wave radars. We focus on FMCW systems [22,23], as these can be made with cheap solid-state transmitters and can measure the range and radial velocity of the targets (hands-gestures). Both Soli and Dop-NET datasets are obtained using this radar technology. Figure 2 details the block diagram of an FMCW radar system, showing a basic radar system with one transmitter and one receiver and data processing modules after the ADC block. In FMCW, the transmit signal is a sinusoid whose frequency is swept linearly from a start frequency $f_c$ to a maximum frequency $f_a$, following a positive slope $\gamma$ and duration of $T_c$, which determines a chirp. The linear frequency modulation (LFM) can have different waveforms, such as triangular, segmented linear, or sawtooth. Figure 3 shows the simplest and most often used sawtooth function, which consists of a series of chirps over

time. The difference between the transmitted and reflected signal from the output of the in-phase mixer, also known as the beat frequency $f_b$ or intermediate frequency (IF), is a new signal that can be measured to determine the distance, velocity, or angle of arrival of reflected objects.

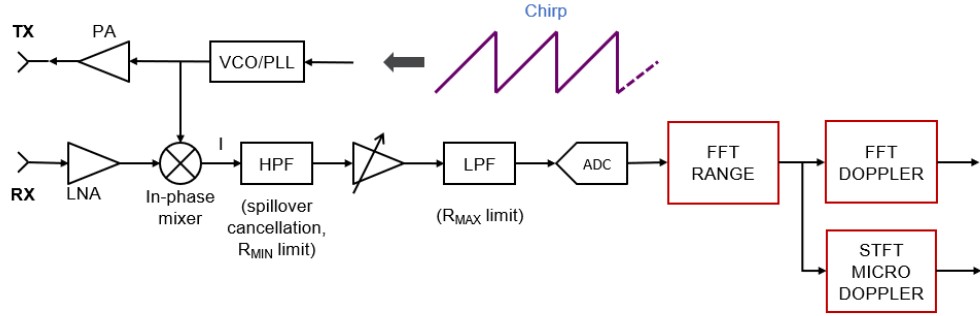

**Figure 2.** Radar block diagram: basic FMCW radar system with one transmitter and one receiver, depicting the typical linear sawtooth waveform and data processing modules after the ADC block.

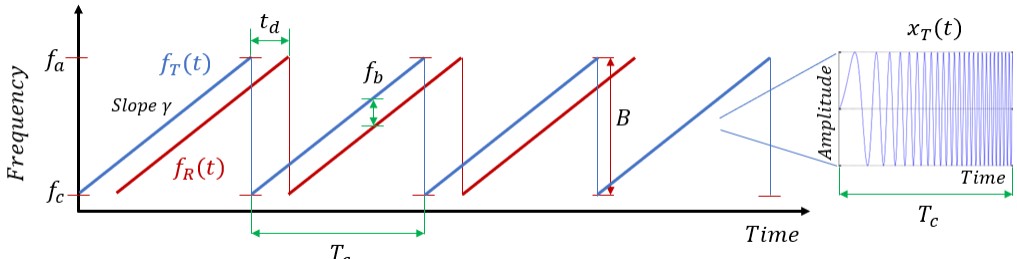

**Figure 3.** FMCW sawtooth wavefor: plot of the transmit (blue) and receive (red) frequency signal over time. $x_T(t)$ indicates the transmit amplitude signal vs time for a chirp interval $T_c$.

Figure 3 depicts the FMCW waveform, showing the plot of the transmit $f_T(t)$ and receive $f_R(t)$ frequency signal over time. The transmitted signal is generated by a modulated continuous wave given by $x_T(t)$, known as the chirp signal:

$$x_T(t) = \cos\left(2\pi f_c t + \pi \frac{B}{T_c} t^2\right) \tag{1}$$

where $\phi_T(t)$ is defined as:

$$\phi_T(t) = 2\pi f_c t + \pi \frac{B}{T_c} t^2 \tag{2}$$

Moreover, the instantaneous frequency is given by:

$$f_T(t) = f_c + \frac{B}{T_c} t \tag{3}$$

where $B = (f_a - f_c)$ is the bandwidth of the chirp.

The received signal is basically the delayed, attenuated version of the transmitted signal:

$$x_R(t) = \alpha x_T(t - t_d) = \alpha \cos\left(2\pi f_c(t - t_d) + \pi \frac{B}{T_c}(t - t_d)^2\right) \tag{4}$$

in which $\alpha$ is the attenuation due to transmission path and $t_d$ is the round-trip delay given by:

$$t_d = \frac{2R}{c} \tag{5}$$

where:

$R$ = range or distance to the object

$c$ = speed of light

Finally, the IF signal after the mixer is a product of the receive and the transmit signals and can be calculated by:

$$y(t) = x_R(t)x_T(t) = \alpha \cos\left(\phi_T(t - t_d)\right)\cos\left(\phi_T(t)\right) \tag{6}$$

$$y(t) = \frac{\alpha}{2}\left[\cos\left(\phi_T(t - t_d) - \phi_T(t)\right) + \cos\left(\phi_T(t - t_d) + \phi_T(t)\right)\right] \tag{7}$$

where it can be shown that the IF signal is proportional to the round-trip delay:

$$y(t) \propto 2\pi\left(\frac{B}{T_c}t_d\right)t \tag{8}$$

and the IF frequency is given by:

$$f_b = \left(\frac{2R}{c}\right)\left(\frac{B}{T_c}\right) \tag{9}$$

After the ADC block, the IF signal is used for object detection, distance (range), and Doppler estimation, typically done by taking fast Fourier transforms (FFTs).

Figure 4 details the pipeline from the ADC block IF signal to the derivation of the range-Doppler or micro-Doppler frames [24]. The IF signal, which follows the typical sawtooth waveform, is composed of chirps, where a frame is defined as a collection of N chirps. For each chirp, data are collected according to a frequency bin size, Figure 4a. In general, an FFT is directly applied for each chirp (fast time), whereas the range bins are stored in the range FFT matrix, Figure 4b. Data from N chirps compose a frame. The range-Doppler frame is created when a second FFT is applied over the chirp indexes (slow time) on all the range FFT bins, Figure 4c. The Doppler profile is centered at zero, corresponding to no movement, whereas positive or negative values indicate motion towards or away from the radar receiver. Micro-Doppler is obtained using a short-time Fourier transform (STFT) on a chirp index over the frames, Figure 4d. The STFT resolution is determined by a window size, which gives a trade-off between the time resolution and the frequency resolution. In general, instead of applying the STFT on a specific chirp index, the index that has the highest value is used for each frame, focusing on the object and creating the micro-Doppler over time spectrogram.

### 2.2. Soli Dataset

The Google Soli project introduced a novel 60 GHz FMCW millimeter-wave radar [3,12,25,26] to create a radar-based sensor optimized for human–computer interaction. The purpose was to develop a sensor for micro-interactions for mobile and wearable computing. A complete end-to-end system was developed, from hardware and software platforms to a machine learning framework, up to the productization by integrating into the Pixel 4 smartphones [27]. Soli is not present in the subsequent releases of the Pixel smartphones. However, Google has recently introduced the second generation Nest Hub, using the Soli radar sensor for sleep tracking [28]. The Soli radar IC is a compact chip, which includes two transmit and four receive antenna elements [12].

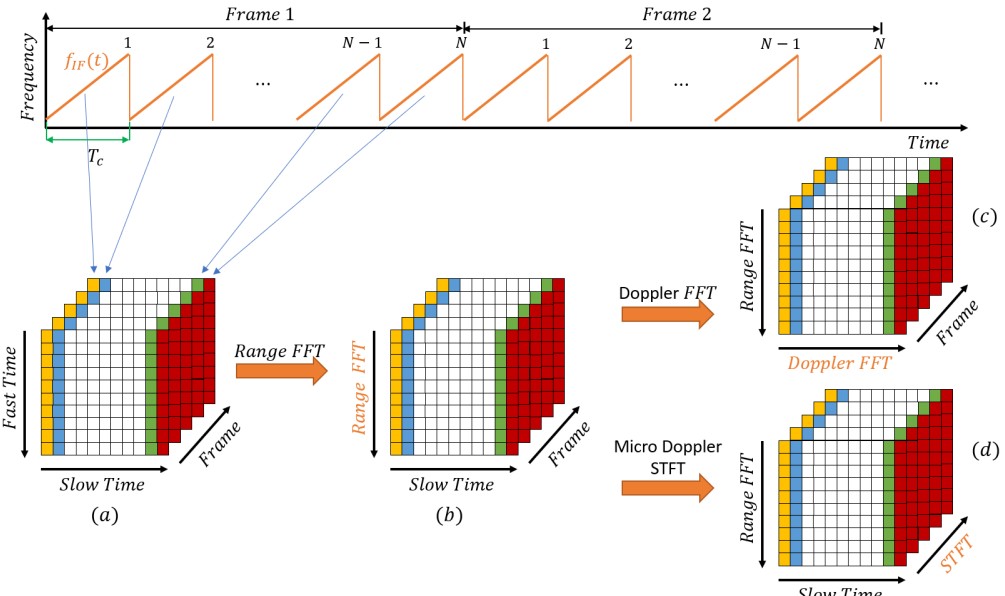

**Figure 4.** FMCW processing flow from the IF signal, assembled in matrix bins. Data are produced according to a frequency bin size (**a**) collected over the chirps. An FFT is directly applied for each chirp (fast time), creating the range FFT matrix (**b**). A second FFT over the chirp indexes (slow time) produces the range-Doppler frames (**c**), while micro-Doppler results from STFT on the chirp index over the frames.

A dataset using the Soli device and a detailed study on hand gesture recognition was presented in [12]. The dataset contains 2D range-Doppler, $32 \times 32$ images, encompassing 11 gestures across ten users, each of which made 25 instances (session) of gestures, totaling 2750 samples. Figure 5a shows an example of the range-Doppler frames, normally obtained following the procedures described in the previous section, where the x-axis represents velocity (cm/ms per unit) and the y-axis distance (cm per unit), see [3]. The range-Doppler images show that per frame data does not convey sufficient information to characterize a whole gesture. Figure 5b depicts a complete data gesture, whereby each frame was unrolled and horizontally stacked, creating a pixel vs. frame matrix. The range-Doppler frames were provided pre-processed using a per-pixel Gaussian model for background removal, and signals were normalized to adjust for variances due to radar reflections. Figure 5c shows the 3D image of the range-Doppler frames, which is equivalent to the matrix representation shown in Figure 4c. It gives a visualization of the reflected radar signal in time, while the Doppler profile shows the distribution of reflected energy over velocity. It is centered in zero, indicating movement toward or away from the radar. For details on the Soli radar signal pre-processing and treatment, see [3,12]. Moreover, for each gesture, the dataset contains four sequences of frames captured by each of the Soli four radar receivers.

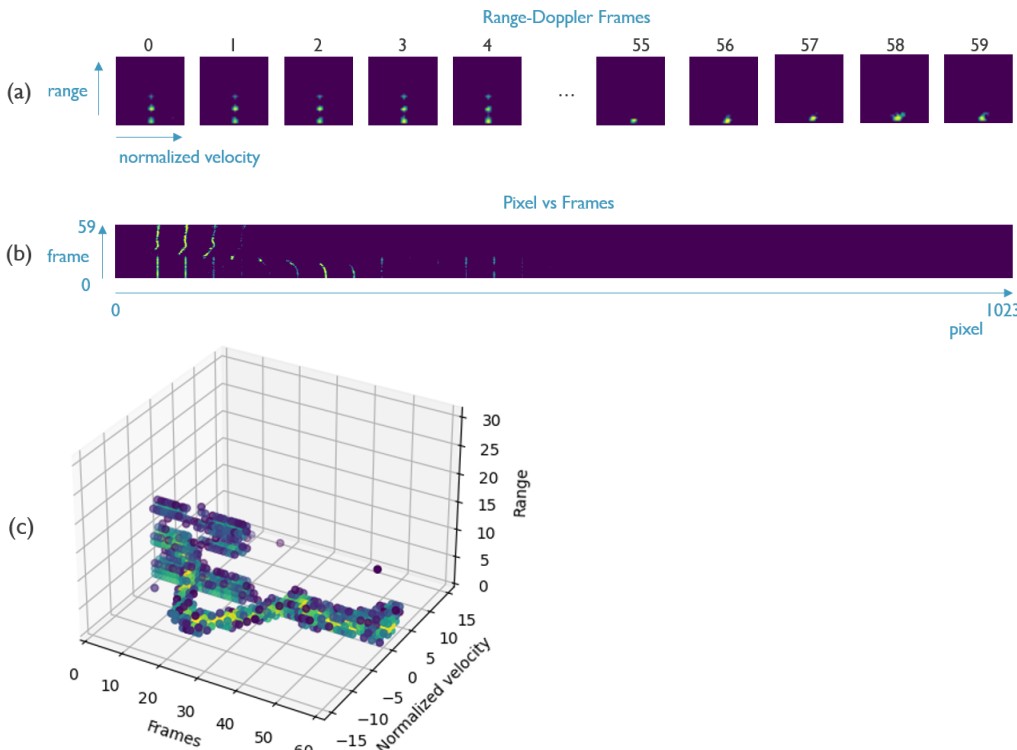

**Figure 5.** Soli Dataset: sample of the range-Doppler frames (**a**), the Doppler information is given as normalized velocity centered at zero. The pixel vs. frame image (**b**) shows the complete gesture capture of one receiver channel. Each frame was unrolled and stacked. For this particular sample, there are 60 frames. The number of frames varies with the time duration of each gesture sample. The 3D image of the range-Doppler frames (**c**) gives a visualization of the reflected radar signal in time, representing a push gesture movement (see Figure 6).

Figure 6 shows all 11 gestures [12] with their respective pixel vs. frame images, clipped to the bottom 384 (32 × 12) pixels. This capping was done because the reflected hand gesture radar signals are mostly located at the bottom position of each 32 × 32 range-Doppler frame, reflecting the range distance from the hands to the radar receiver when performing the hand gesture experiments. Consequently, we could reduce and optimize the data representation, see more detail in Section 2.4. Moreover, the pixel vs. frame representation captures the information in time, which is favorable for the signal-to-spike neural encoding to produce the spike train input for the LSM network. Figure 6 also shows 3D visualization of four gestures, showing the radar signal representation of the gesture through time.

### 2.3. Dop-NET Dataset

The Dop-NET dataset was introduced in [16], while it describes measurements from two types of radars, a continuous wave (CW) and FMCW, across ten different people, only the 24 GHz, FMCW radar data from six people, was released. Moreover, the radar had one transmit and two receive antennas, but the dataset included only the co-polarized channel data. The gestures were made, in front, at the same height of the radar, and at around 30 cm distances, creating a relatively uniform gesture localization on the spectrograms. Unlike the Soli dataset, the radar signal in Dop-NET is based on micro-Doppler signatures. Moreover, the data are provided as a matrix of complex values representing the Doppler vs. time for each gesture. Section 2.1 details the procedures on how micro-Doppler is obtained from an FMWC radar system.

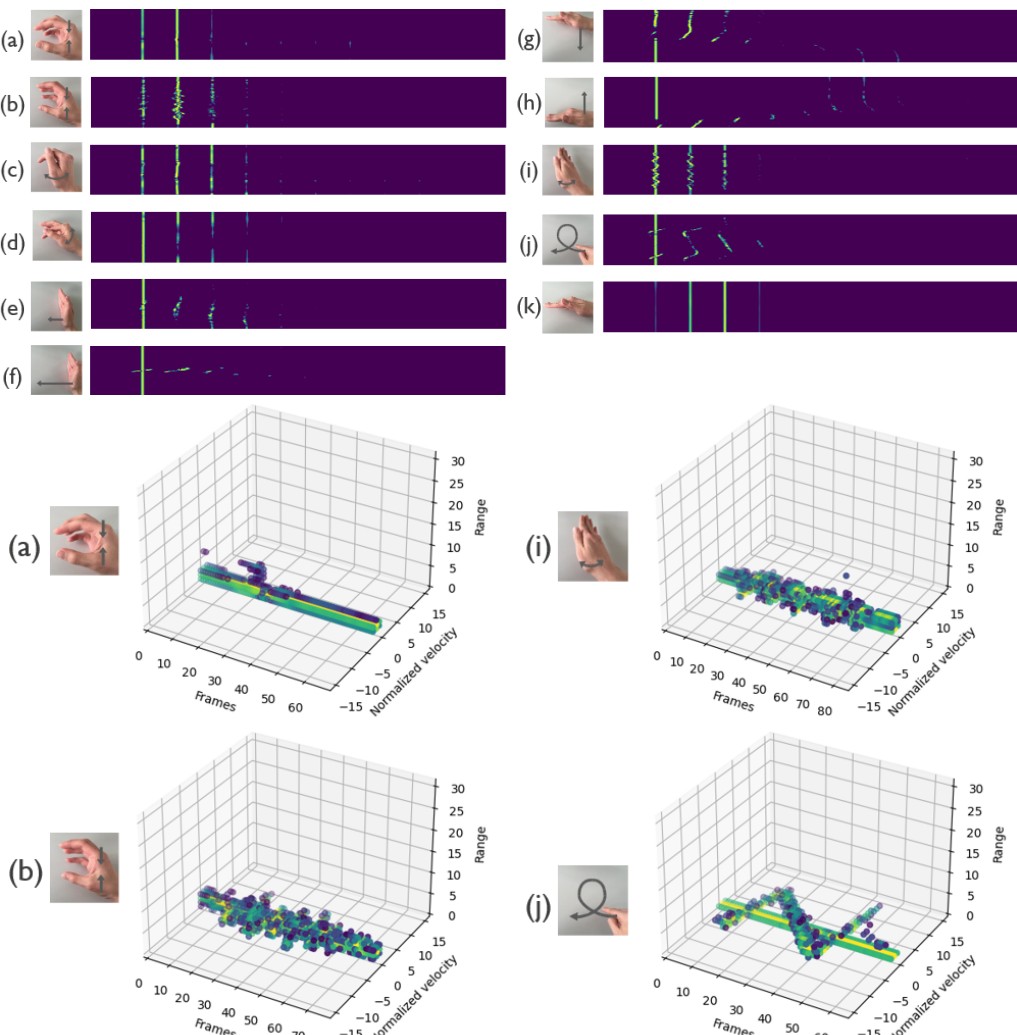

**Figure 6.** Soli Gestures: sample of 11 gestures with their respective pixel vs. frame capped image. (**a**) Pinch Index, (**b**) Pinch Pinky, (**c**) Finger Slide, (**d**) Finger Rub, (**e**) Slow Swipe, (**f**) Fast Swipe, (**g**) Push, (**h**) Pull, (**i**) Palm Tilt, (**j**) Circle, (**k**) Palm Hold. Figures at the lower part show the 3D image of the range-Doppler frames for four gestures.

Figure 7 shows a sample of the spectrograms of the four gestures, which consists of the amplitude of Doppler over time. Thus, phase information was not used when composing the data representation as input for the recognition system. The signal to spike encoding scheme is detailed in the next Section 2.4.

In Dop-NET, each gesture can have a distinct duration reflected in the different time axis length, which varies from 43 to 540 steps (for Soli, this is expressed in the number of frames in each sample). Dop-NET's main goal was to provide real radar data for the research community as a data challenge with a specific test dataset as in a competition. The complete dataset contains four gestures from six users, with a separate test dataset (see Table 1).

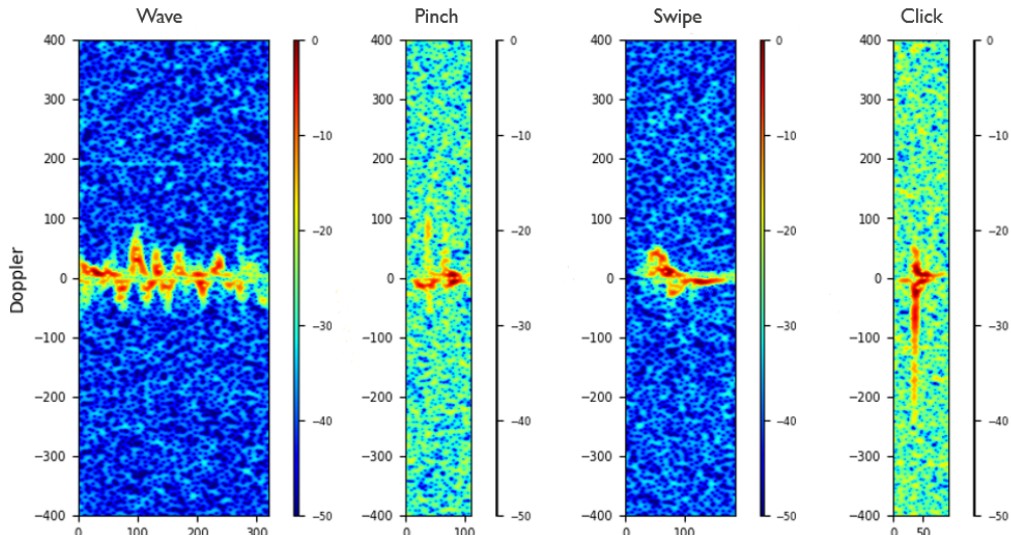

**Figure 7.** Dop-NET: Micro-Doppler time vs. frequency spectrograms of the 4 gestures (Wave, Pinch, Swipe and Click).

**Table 1.** Dop-NET: Number of samples in the training and test data.

| Training Data | | | | |
|---|---|---|---|---|
| **Person** | **Wave** | **Pinch** | **Swipe** | **Click** |
| A | 56 | 98 | 64 | 105 |
| B | 112 | 116 | 72 | 105 |
| C | 85 | 132 | 80 | 137 |
| D | 70 | 112 | 71 | 93 |
| E | 56 | 98 | 91 | 144 |
| F | 87 | 140 | 101 | 208 |
| **Test Data** | | | | |
| | | 643 | | |

### 2.4. Data and Spike Train Representation

We used two distinct datasets based on different radar signals to address radar-based HGR more generically. While the Soli dataset is based on range-Doppler (Section 2.2), Dop-NET provides Doppler vs. time signatures (Section 2.3), leading us to two distinct initial pre-processing steps, however, with a common data-to-spike representation and spike train encoding scheme.

The spike train encoding for the Soli dataset is based on the capped pixel vs. frame introduced in Section 2.2. In the first step, the range-Doppler signal can be binarized using a simple threshold function, thus given a signal $f_i(t)$, whereas $i$ index the signal at pixel position $i$:

$$P_i(t) = \begin{cases} 1, & \text{if } f_i(t) > \theta \\ 0, & \text{otherwise} \end{cases} \tag{10}$$

where $\theta$ denotes a threshold value.

For Soli, as mentioned, the radar range-Doppler frames were provided pre-processed with a per-pixel Gaussian model, and signals were normalized to adjust for radar reflections. Since the radar signal was filtered against noise and normalized, accounting for the variability, e.g., hand size, speed, and distance using $\theta = 0$, to convert the signal to spikes captured the gesture representation. The threshold $\theta$ can play a role as a noise filter or control the amount of sparsity in the representation. In our case, it was straightforward to

transform the radar signal in time to a set of spike trains representing the gesture and used as input into the LSM.

For each sample the spike train from pixel position $i$ is given by a vector: $s_{it_0}^T$ where $t_0$ is the initial spike train time and $T$ is the final time, and $s_i(t_0), s_i(t_0 + 1), s_i(t_0 + 2), \ldots, s_i(T)$ are the individual spike events. From the binarized frames, each spike at frame $n$ is translated to a time $s_i(t)$, relative to the length of the particular gesture, see Section 2.5.

The micro-Doppler spectrograms from the Dop-NET (see Figure 7) required a different pre-processing step in contrast to the simple threshold (Equation (10)) used in Soli. We have experimented with different techniques, from biological neural-inspired methods, to analyze the spectrogram as an image problem. The objective was to isolate the signal from the spectrogram that yields the best information in time characterizing the gestures. The best approach used image processing techniques. First, we converted the spectrogram signal to a grayscale (0–255) image, then applied a Gaussian filter, and the Minimum Error Thresholding (MET) [29,30]. The Gaussian filter removes noise and smothers the edges on the images [31], while the minimum error thresholding binarizes the image. Thresholding can be seen as a segmentation problem where the objective is to classify each pixel as an object or background. MET is a recursive clustering algorithm that minimizes the misclassification error. It improves over the well-known Otsu's method [32], and it is shown to be one of the best image thresholding methods [33].

Figure 8 shows the same samples from Figure 7, capped on the y-axis to the central 400 pixels, after binarization. The x-axis represents time, while the spike train will contain the spike times at pixel position $i$, in the same way as used in the Soli dataset, whereas the set of spike trains, $s_{it_0}^T$, represents a gesture sample and are the input for the LSM.

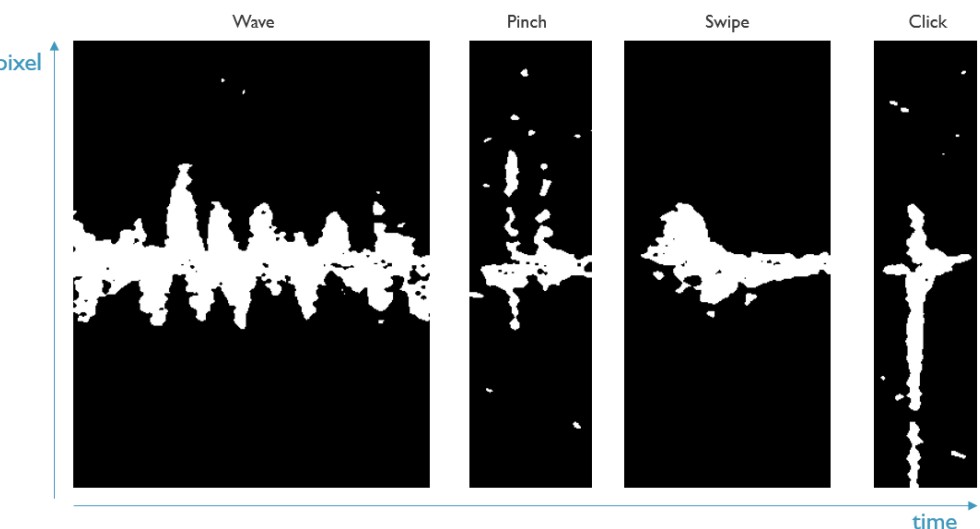

**Figure 8.** Wave, Pinch, Swipe, and Click binarized from the micro-Doppler spectrograms. Thus, the spectogram was converted to a grayscale image, followed by a Gaussian filter, and the minimum error thresholding.

### 2.5. Liquid State Machine

A Liquid State Machine (LSM) is a type of reservoir computer capable of universal function approximation [17,18]. The basic formulation of LSM maps an input function $u(\cdot)$ onto a filter, or liquid neurons, $L^M$ while the output $x^M(t) = (L^M u)(t)$ is fed to a second component, a readout map $f^M$, which is task-specific and generates the output $y(t) = f^M(x^M(t))$. The readout maps in our context will be a classifier that receives a state as input. We used three different classifiers for comparison, i.e., logistic regression, random forest, and support vector machine, thus from linear to ensemble-based methods, showing the effectiveness of the second component for the overall hand gesture classification task. The liquid $L^M$ is composed of excitatory ($E$), and inhibitory ($I$) neurons, with 20% being configured as inhibitory, using the same heuristic as in [19]. The synaptic connections were

randomly assigned, creating a sparse network with the following ratios $EE = 2$, $EI = 2$, $IE = 1$, and $II = 1$, where $EE$ denotes the number of synapses connecting excitatory ($E$) to excitatory ($E$) neurons, $EI$ excitatory ($E$) to inhibitory ($I$) and so on. These ratios will ensure that the connection density is not very high, as some spiking network tends to be chaotic with high connection density [18].

Figure 9 shows the LSM and how it has been used to build an end-to-end system for gesture recognition. The previous sections have described how, from the range-Doppler or micro-Doppler signals, we created the spike trains that characterize each gesture sample. The top part at Figure 9 depicts the timing and how each gesture is sampled in the LSM. $T_{s_0}$ and $T_{s_1}$ are the boundaries of the time interval reserved for a gesture, wherein the spike train of a gesture can have a variable stimulus length duration. After the end of the stimulus, a readout delay $t_r$ determines the readout window interval, during which the state of the liquid is measured and stored or passed to the classifier, depending on if used on a real-time online or offline learning and inference system.

Each gesture sample had a different time duration. In the case of Soli, the samples varied from 28 to 145 frames. When mapping to the LSM, each sample had a different stimulus length. As a result, the readout window varies according to the sample frame duration. The conversion from spike at pixel position $i$, of frame $n$ to spike $s_i(t)$ at time $t$, is a direct map from $n$ to $t$, i.e., if frame $n$ has a spike at pixel $i$, then $s_i(t)$ has a spike at $t = n$.

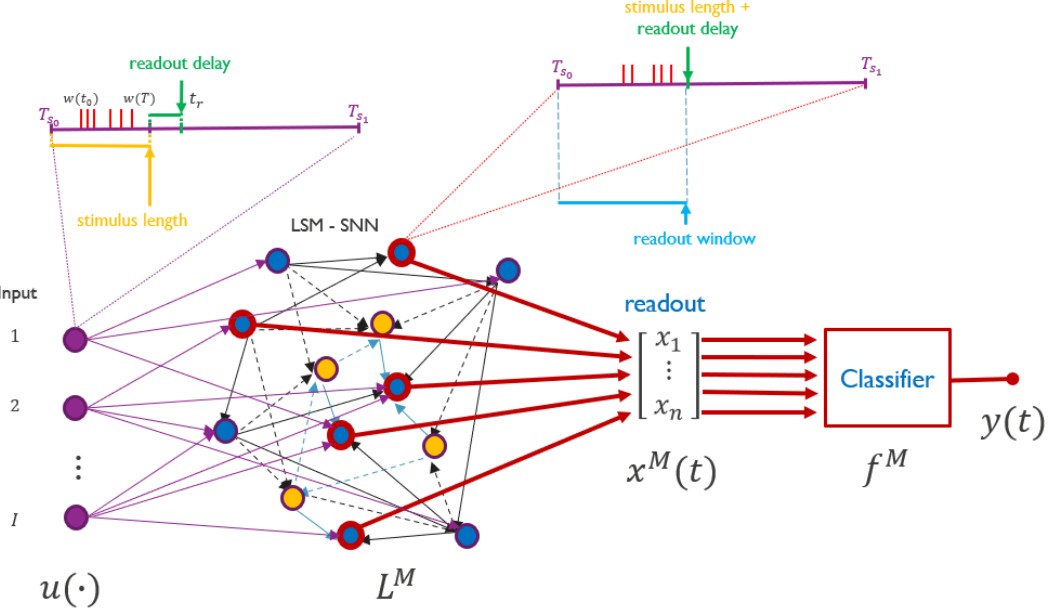

**Figure 9.** Schematic of the LSM given by $x^M(t) = (L^M u)(t)$ and $y(t) = f^M(x^M(t))$. $T_{s_0}$ and $T_{s_1}$ are the initial and end time interval a gesture is sampled in the LSM. While $s_i(t_0), s_i(t_0 + 1), s_i(t_0 + 2), \ldots, s_i(T)$ are the individual spike events at input $i$. $t_r$ is the readout delay, and the readout windows gives the period which measures the state of the liquid, that is used as input to the classifiers.

An alternative way to map the LSM was to normalize the frames to a predefined fixed stimulus length, whereas all the samples have the same readout window duration. For every pixel position $i$, we convert the spike in frame $n$ to a relative time regarding a fixed stimulus length $S_l$. Thus, the spike train sequence $s_{i t_0}^T$ is given by:

$$s_i(t) = \frac{f_n * S_l}{f_l} \tag{11}$$

where:

$S_l$ = predefined fixed stimulus length;

$f_n$ = frame number, $n$ that contains a spike;

$f_l$ = length of the particular sample in number of frames.

Section 3 shows that this normalization results in about 0.5% to 2% better accuracy than treating the gesture signals as a real-time flow of events. In constructing the LSM, we focus on achieving the most compact and simpler to implement network without sacrificing high accuracy results. Each pixel $i$ will produce a spike train as an input to the LSM, and each input is randomly connected to $C_{inp}$ excitatory neurons. Moreover, all excitatory neurons are used for readout. For the neuron unit, we used a leaky integrate-and-fire neuron model with exponential postsynaptic currents with the associated synaptic model, based on [34]. However, while we tested synaptic models incorporating plasticity, a straightforward static synapse with a limited range of weight values was sufficient to achieve high accuracy results on Soli and Dop-NET datasets.

## 3. Experiments and Results

This section shows the results of applying the proposed signal to spike neural encoding and LSM architecture previously described in Section 2 to the Soli and Dop-NET data sets. Each of the datasets was evaluated separately, with their corresponding gesture sets, i.e., 11 gestures for soli and four gestures in Dop-NET. Besides specific tests described in the benchmark papers, we analyze three cases:

- K-fold cross-validation: partitioning the complete data set in complementary k-subsets using k-1 subsets for training and one subset for testing, repeat k time for cross-validation.
- Leave-one-subject-out cross-validation: training the system on all but one participant and testing the remaining participant. It indicates how well a pre-trained system on several users would perform for unknown users.
- Leave-one-session-out cross-validation: training the system for a particular user only, using all but one of the user's sessions to train, and testing on the remaining one. It indicates how well the system could be used on a personal device trained for just one particular user.

Furthermore, we examine how the different parameters contribute to the spiking neural network and the overall classification results. For the simulations, we explore the impact of the size of the network and the input stream $u(\cdot)$ on the accuracy results of the system. While testing on LSM with varying numbers of neurons, we kept the ratio of excitatory and inhibitory neurons and the synaptic connectivity as described in Section 2.5. Moreover, each input signal $i$ is connected to $C_{inp}$ neurons, which determines the amount of signal transmitted to the liquid. Generally, the external stimuli are sent to a fixed percentage of the neurons, always chosen to be excitatory [20]. However, we vary $C_{inp}$, modifying this ratio to determine its impact on the network.

Using three different classifiers for comparison, we aimed to keep the parametrization of the classifiers as common and straightforward as possible, which were applicable for both datasets. We used a logistic regression model with a regularization configuration of $C = 1$, using L2 penalty, and allowed 5000 iterations for convergence. For random forest, we parametrized the ensemble to grow to 200 trees and used entropy as a metric for feature importance evaluation, i.e., quality of the decision split. Finally, for SVM, we used a radial basis function kernel with L2 penalty and a high $C = 128$ value parameter, aiming for a better training set classification. In other words, it considered points closer to the line of separation between the classes. Alternatively, having a weaker penalty (lower $C$) caused higher misclassification. The C parameter here is defined as in scikit-learn [35], whereas the strength of the regularization is inversely proportional to C.

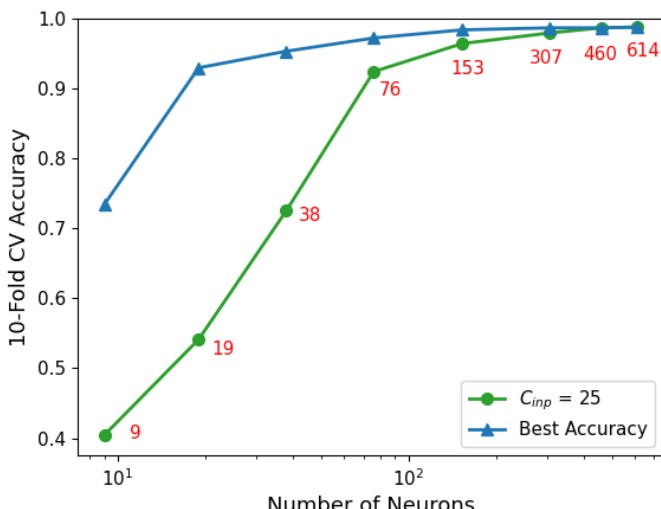

**Figure 10.** The 10-fold cross-validation accuracy vs. LSM size. Red numbers indicate the exact number of neurons in the network. Green curves follows the accuracy at a fix $C_{inp} = 25$, while the blue curve gives the best accuracy irrespective of $C_{inp}$ for the given LSM size.

### 3.1. Soli

This section discusses the results for Soli, where we used the signal of just one of the radar receivers, i.e., channel 3. Figure 10 shows accuracy vs. size of the network, i.e., number of neurons, for 10-fold cross-validation, using an SVM classifier. The numbers in red indicate the total number of neurons in the LSM for each of the experiments. The blue line shows the best accuracy for the given LSM size (evaluated over different values for $C_{inp}$). It shows that with a relatively small network (153 neurons), it is already possible to achieve high accuracy of 98.3% (blue), while 98.6% was the best overall result using 460 neurons and $C_{inp} = 25$ (see Table 2). The green curve shows accuracy at a fix $C_{inp} = 25$, for different network sizes.

**Table 2.** Accuracy result comparison with Soli. Results for 50–50% split for training and evaluation, 10-fold cross-validation, leave one subject and session out, using an SVM classifier. For an LSM with 460 neurons and $C_{inp} = 25$.

|  | Soli | Normalized $S_l$ | Variable $S_l$ |
|---|---|---|---|
| Training and evaluation | 87.17% | 98.02% | 97.51% |
| 10-fold cross-validation |  | $98.6 \pm 0.7\%$ | $97.8 \pm 0.9\%$ |
| Leave-one-subject-out | 88.27% | $94.2 \pm 3.5\%$ | $91.4 \pm 3.8\%$ |
| Leave-one-session-out | 94.15% | $98.8 \pm 0.8\%$ | $97.4 \pm 1.4\%$ |

Figure 11 shows the 10-fold cross-validation accuracy vs. LSM size for all three classifiers. It shows the best results irrespective of $C_{inp}$, indicating that SVM was the best of the three classifiers. Moreover, as the size of the network increases, the best accuracy for all three classifiers tends to achieve a high value. This happens as a consequence of the LSM separation property, as increasing the neural circuit increases the separation property of the liquid as demonstrated in [19].

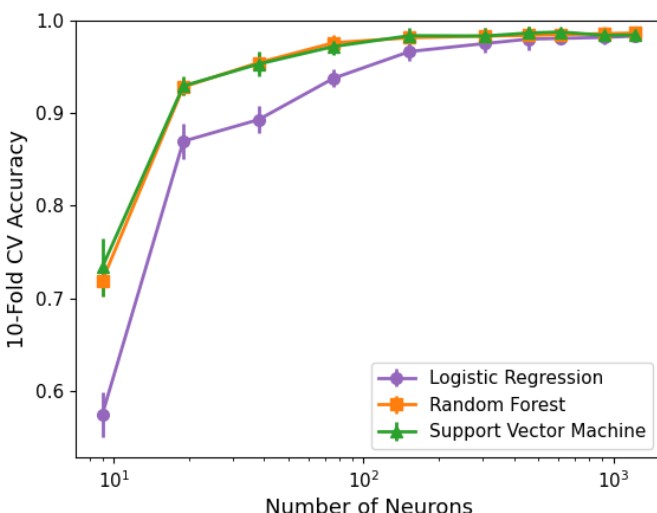

**Figure 11.** The 10-fold cross-validation accuracy vs. LSM size for all three classifiers. The error bars indicate the standard deviation. It shows the best results irrespective of $C_{inp}$.

Figure 12 shows the results for 10-fold cross-validation, on an LSM with 460 neurons, using three different classifiers. The SVM classifier gives the best result of 98.6%, but even the logistic regression reaches near to 98% accuracy. As discussed in Section 2.5, there are two alternatives to encode the temporal signal of the gestures. The graphs on the left show the accuracy results when the spike trains are normalized to a common stimulus length. In this case, we used $S_l = 50$, which is around the average temporal length of the gestures in the Soli dataset. On the right are the results when $S_l$ varies according to the number of frames in each sample gesture. It is shown that, by normalizing to a reference stimulus length, we have slightly better accuracy results.

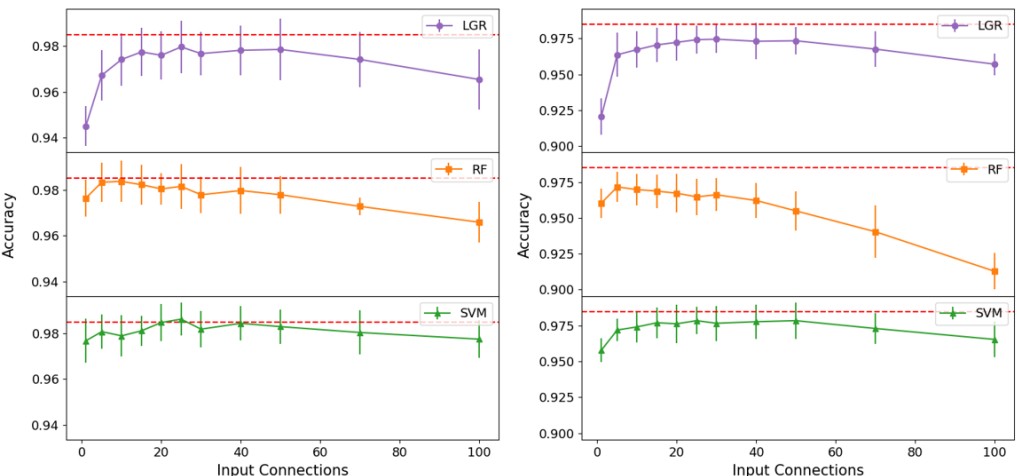

**Figure 12.** Soli: graphs of accuracy vs. varying $C_{inp}$ using 10-fold cross-validation. Results are for an LSM with 460 neurons using logistic regression, random forest, and support vector machine. The red line indicates accuracy at 98.5%. The left graph shows the results when the spike time of each sample is normalized to a common stimulus length. While the right graph when each sample has a stimulus length equivalent to the sample frame duration.

Figure 12 also shows how accuracy varies with an increasing $C_{inp}$ at an LSM with a fixed size. Independent of the classifier, accuracy generally increases and rapidly achieves a plateau, then goes down with an increasing $C_{inp}$. We conjecture that, as $C_{inp}$ increases, it also increases the amount of signal transmitted to the liquid. The network will be

dominated by the external stimuli, degrading the contribution from the recurrent network and impacting the fading memory capability of the LSM, which is a fundamental property for the universal computational power of the LSM [18,19]. As the size of the LSM increases, the plateau shifts, indicating that an optimal combination of LSM size and $C_{inp}$ exist to achieve a high accuracy rate and minimal LSM size. Our objective was to achieve the smallest and simple network that can achieve acceptable accuracy rates.

The Soli dataset also contains a file indicating which samples were used in a 50–50% split for training and evaluation. Using this file, we applied an LSM with 460 neurons, $C_{inp} = 25$, and an SVM as a classifier resulting in the confusion matrix presented in Figure 13. The table on the left shows the results using a normalized stimulus length, while the right confusion matrix details the results when using the stimulus length according to the time length of each gesture sample.

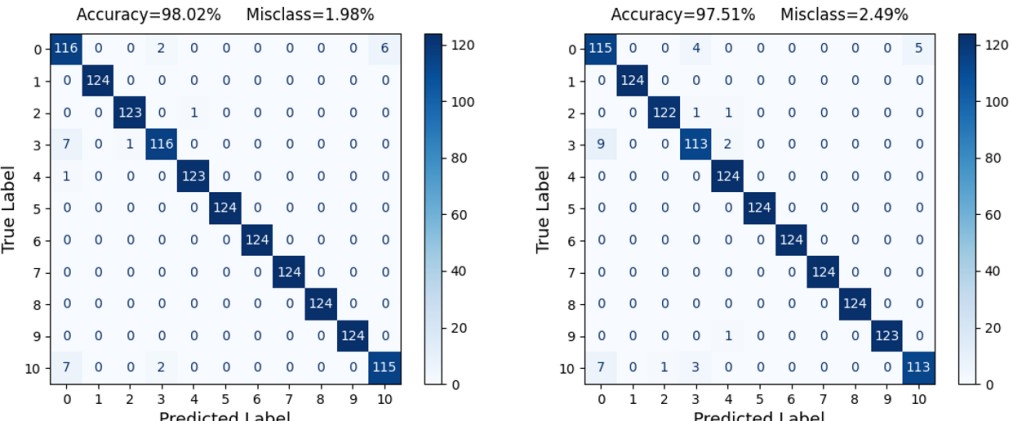

**Figure 13.** Confusion matrix for the Soli 50–50% split for training and evaluation. Results for an LSM with 460 neurons, $C_{inp} = 25$ and using SVM classifier. Left table shows the results for normalized $S_l$ and the right for variable $S_l$.

Table 2 shows the main results, putting them side by side with the results from [12], which includes the training and evaluation split and the Soli results for leave-one-out cross-validations with sequence average-pooled accuracy. We achieve better results, using a much smaller network on all the cases. Moreover, we perform 10-fold cross-validation on the whole dataset, which gives better confidence to the results given the dataset. Table 2 shows that the leave-one-subject-out cross-validation has a higher standard deviation, which is to be expected as some personalization to the gestures exists, and the training dataset limits the learning generalization. Finally, the leave-one-session-out result shows that learning for a personalized use case can achieve high results with a relatively limited learning set.

### 3.2. Dop-NET

Figure 14 shows the same interplay between the size of the network, $C_{inp}$ and accuracy, as in the Soli experiments. In the Dop-NET case, the green curve indicates accuracy results when $C_{inp} = 5$, while the blue shows the best accuracy irrespective of $C_{inp}$, for the LSM sizes indicated in red. Again depending on an acceptable accuracy rate, a relatively small network (240 neurons) is capable of achieving high accuracy depending on $C_{inp}$, 97.63% (blue). The best overall result is 98.60% using 960 neurons.

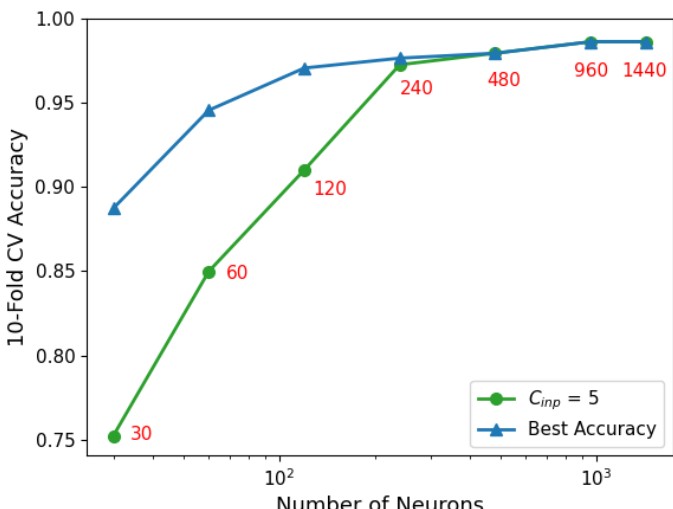

**Figure 14.** The 10-fold cross-validation accuracy vs. LSM size. Red numbers indicate the exact number of neurons in the network. Green curves follow the accuracy at a fix $C_{inp} = 5$. The blue curve gives the best accuracy irrespective of $C_{inp}$ for the given LSM size.

Figure 15 shows the best 10-fold cross-validation accuracy results irrespective of $C_{inp}$ over different LSM sizes for all three classifiers. It shows a similar behavior as in Figure 11 for Soli, as the size of the network increases, the separation property of the liquid increases, and the classifiers are able to achieve equivalent results.

Figure 16 shows the equivalent results for Dop-NET, using 10-fold cross-validation, on an LSM with 960 neurons. In this case, the effect of increasing $C_{inp}$ is more pronounced, whereas there is hardly a plateau and accuracy peaks at a low $C_{inp}$ value, steadily decreasing with increasing $C_{inp}$, in the same way, as discussed previously in the Soli case. Once more, this indicates that an optimal combination of LSM size and $C_{inp}$ for minimal LSM size, capable of high accuracy rates. The left graph shows the results for normalized stimulus length at $S_l = 200$, which is around the average time length of the gesture samples. On the right, it shows the results where $S_l$ varies according to each sample.

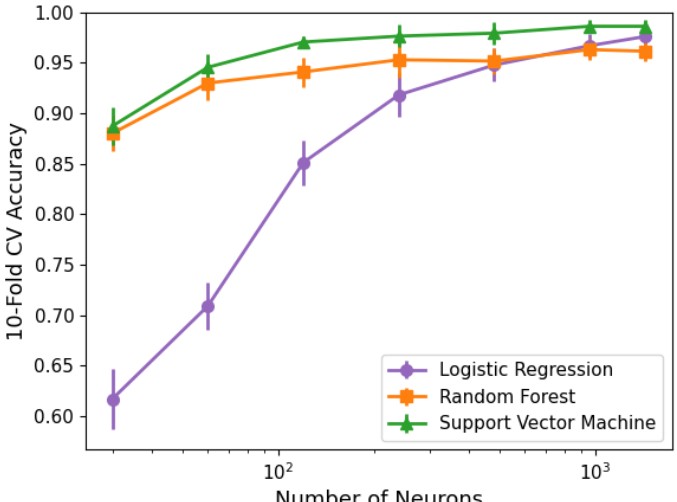

**Figure 15.** The 10-fold cross-validation accuracy vs. LSM size for all three classifiers. The error bars indicate the standard deviation. It shows the best results irrespective of $C_{inp}$.

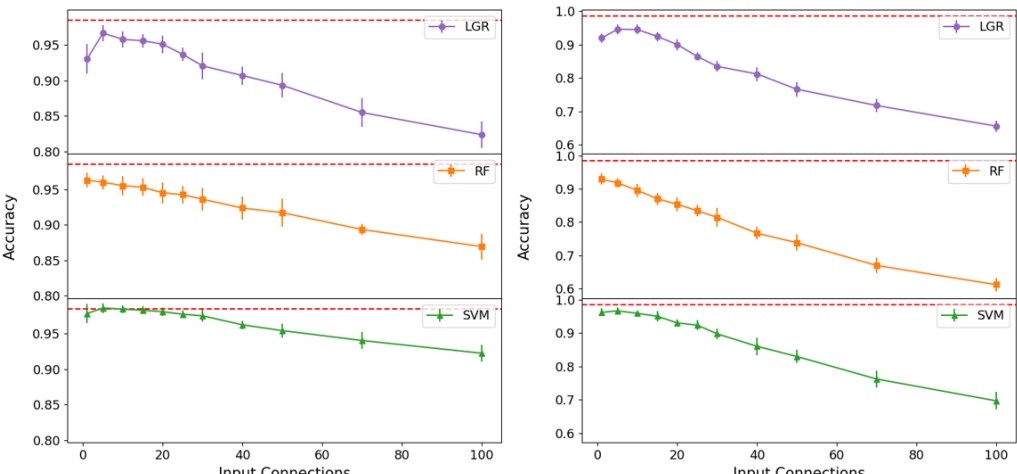

**Figure 16.** Dop-NET: Graphs of the accuracy vs. varying $C_{inp}$ using 10-fold cross-validation. Results are for an LSM with 960 neurons using logistic regression, random forest, and support vector machine. The red line indicates accuracy at 98.5%. The left graph shows the results when the spike time of each sample is normalized to a common stimulus length. While the right graphic when each sample has a stimulus length equivalent to the sample time duration.

The Dop-NET dataset was initially conceptualized as a data challenge, with specific training, and test data files. Table 1 details the training dataset; notice that the number of samples is not uniform, neither across persons nor gestures. The test dataset was provided separately, initially without the labels, which were eventually made available. Figure 17 shows the confusion matrix for both using normalized $S_l$ and variable $S_l$. We demonstrate that our system would achieve very high accuracy on the challenge.

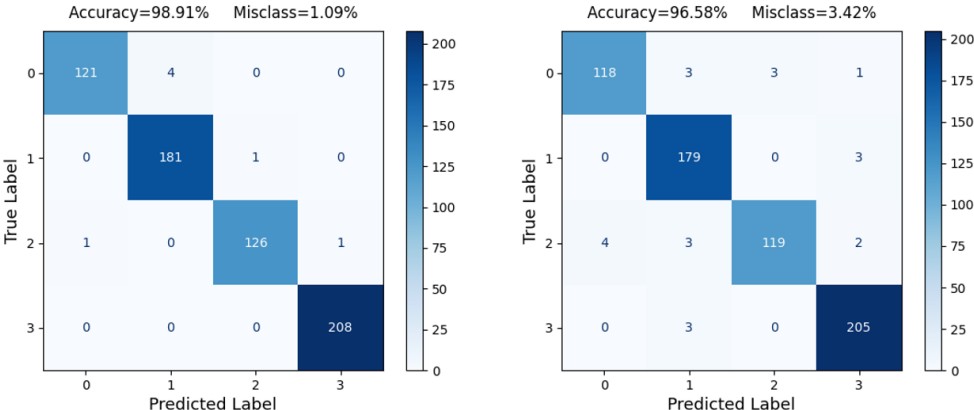

**Figure 17.** Confusion matrix for the Dop-NET using the training dataset and evaluated on the specific test set. Results for an LSM with 960 neurons, $C_{inp} = 5$ and using SVM classifier. The left table shows the results for normalized $S_l$ and the right for variable $S_l$.

Table 3 lists the main cross-validation results. A direct comparison, using the training and test sets, can be made with the results from [16], whereas they used five features extracted from the micro-Doppler spectrograms and achieved the best result of 74.2% accuracy, using SVM as a classifier. In other experimental results using Dop-NET [2], the authors compare results from co-polarized and cross-polarized radar signals and using single or multiple range bins. As mentioned in Section 2.3, our dataset contains only co-polarized data from a single range bin. In this category, they achieved the best result of 89%, using a K Nearest Neighbor (KNN) classifier, with a 90–10% split of training and test data. We achieve a better result of 98.6% on 10-fold cross-validation. Our results for the leave-one-subject-out cross-validation are shown to be relatively low. We attribute this to one of the persons having a different signature than the others in the dataset as indicated

in [2]. Finally, for the leave-one-session-out, we assumed a maximum of 50 sessions for each user. As the minimal number of samples of a specific gesture was 56, for the wave gesture for persons A and E, see Table 1. Moreover, having the same amount of sample for all the users and gestures gives a more uniform comparison. We achieved a very high accuracy rate, showing that a personalized learning system can be achieved with limited data.

**Table 3.** Dop-NET results for an LSM with 960 neurons and $C_{inp} = 5$, using SVM as a classifier. Lists of the results when using the specific training and test sets for comparison with [16], the 10-fold cross-validation, and the leave-one-subject-out and leave-one-session-out.

|  | **Dop-NET** | **Normalized $S_l$** | **Variable $S_l$** |
|---|---|---|---|
| Training–Testing | 74.20% | 98.91% | 96.58% |
| 10-fold cross-validation |  | $98.6 \pm 0.6\%$ | $96.6 \pm 1.0\%$ |
| Leave-one-subject-out |  | $69.6 \pm 20.1\%$ | $65.5 \pm 17.9\%$ |
| Leave-one-session-out |  | $99.1 \pm 0.8\%$ | $97.5 \pm 1.3\%$ |

### 3.3. State-of-the-Art Radar-Based Gesture Recognition

Table 4 reports deep neural networks and machine learning models presented in literature solving the radar-based gesture recognition task on the Dop-NET and Soli datasets. We report the best accuracy to highlight the superior performance of the proposed spiking recurrent neural network model of the present work.

The spatio-temporal nature of the gesture recognition task mainly roots the increase in performance of the spiking neural network. The high-dimensionality expansion performed by the liquid makes the training of the readout units more effective, as shown in Figures 12 and 16.

**Table 4.** State-of-the-art radar-based gesture recognition.

| Network/Algorithm | Best Accuracy | Dataset | Reference |
|---|---|---|---|
| Spiking LSM | **98.02**% | Soli | this work |
| CNN + LSTM | 87.17% | Soli | [12] |
| Random Forest | 92.1% | Soli | [3] |
| Spiking LSM | **98.91**% | Dop-NET | this work |
| SVM Quadratic | 74.2% | Dop-NET | [16] |
| K Nearest Neighbor | 87.0% | Dop-NET | [2] |

## 4. Discussion

One of the main focuses of our research is to find the most compact and straightforward neuromorphic systems capable of solving challenging spatio-temporal pattern recognition tasks. A radar-based recognition system is an actual and challenging task that can be used in many applications. We have shown the possibility of achieving a relatively small (1000 units) SNN system capable of solving radar-based HGR more accurately than using other machine learning and deep learning approaches. In addition, its simple training mechanism allows building personalized systems with very high recognition accuracy. At the same time, there is much to be explored further, including automatic ways to jointly optimize the size, input connectivity $C_{inp}$, and readout window of the system, mechanisms for self-learning liquid state machines, tightly concatenated or overlapping gesture recognition, and online learning of new gestures to expand the overall system capabilities.

Our approach is compatible with state-of-the-art neuromorphic systems. In fact, many neuromorphic spike-based systems are composed of mixed-signal designs in which neurons and synapses are implemented using sub-threshold analog circuits. These have been proposed for their low power performance and their bio-realistic synaptic and neuronal dynamics [36–39]. However, these systems suffer from noise, variability, and mismatch.

A solution to the variability and mismatch problems is partially provided when using an LSM network, requiring only a high-precision readout layer. The LSM network with dynamic synapses, like the one proposed in this work, is set up with randomly connected recurrent weights to project the input signal to a higher-dimensional space. The neuron's and synaptic's dynamic, and the high dimensional space, provide a basic pool of functions [19]. The projected spatiotemporal stream of data (i.e., the spike trains) can then be separated because of the Kernel properties of the LSM [40]. This step only requires training the readout layers on a buffered stream (readout window) of outputs with high-precision synaptic weights. These weights and this single matrix multiplication can be implemented in digital hardware logic. At the same time, the majority of the neurons, with their dynamical synapses, can reside in mixed-signal low-power hardware. This approach has already been demonstrated for the case of Electromyography (EMG) gesture recognition [41] and heartbeat classification [42]. The only downside of this approach is the requirement of the synchronization of the sequence (one particular hand gesture or heartbeat) within a readout output window. In the present work, we have used the NEST simulator [43,44] for simulating bio-realistic neuronal and synaptic dynamics. Still, a natural extension is the implementation of the network in mixed-signal neuromorphic hardware.

**Author Contributions:** Conceptualization, all authors; investigation, I.J.T. and W.V.L.; software, I.J.T.; writing, I.J.T., W.V.L. and F.C.; writing—review, M.S. and S.L. All authors have read and agreed to the published version of the manuscript.

**Funding:** This research received no external funding.

**Data Availability Statement:** The data presented in this study are openly available in Soli https://github.com/simonwsw/deep-soli (accessed on 30 April 2021) and Dop-NET https://dop-net.com (accessed on 30 April 2021). The code will be made available upon request.

**Acknowledgments:** The resources and services used in this work were provided by the VSC (Flemish Supercomputer Center), funded by the Research Foundation—Flanders (FWO) and the Flemish Government.

**Conflicts of Interest:** The authors declare no conflict of interest.

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
