# Peer review of "Radar-Based Hand Gesture Recognition Using Spiking Neural Networks"

_electronics, doi:10.3390/electronics10121405_

Round 1

Reviewer 1 Report

In this paper a radar-based hand gesture technique by exploiting neural networks has been presented.

Although the article has merit, the radar theory and operation and the description of the paper’s novel elements should be improved.

The absence of privacy concerns could be cited as an advantage of radars over the camera-based systems.

Fig. 2 does not transmit any messages without describing what is the gesture being measured. From the range Doppler graph seems that there are not Doppler movements because the points are always centered in zero (if the graph goes from negative to positive Doppler frequencies, according to the classical representation).

I understand that the paper is mainly focused on the neural network description, but this is a journal speaking about electronics thus the radar theoretical description must be more solid.

To this aim, the authors can find useful recent references, hereafter, where interesting related information and details can be found also for the readers.

  • - H. Li, A. Mehul, J. Le Kernec, S. Z. Gurbuz and F. Fioranelli, "Sequential Human Gait Classification With Distributed Radar Sensor Fusion," in IEEE Sensors Journal, vol. 21, no. 6, pp. 7590-7603, 15 March15, 2021, doi: 10.1109/JSEN.2020.3046991.
    - E. Cardillo, C. Li and A. Caddemi, "Vital Sign Detection and Radar Self-Motion Cancellation Through Clutter Identification," in IEEE Transactions on Microwave Theory and Techniques, vol. 69, no. 3, pp. 1932-1942, March 2021, doi: 10.1109/TMTT.2021.3049514.

This is a contribution focused on the radar features thus I strongly recommend to expand the radar description, both theoretically and from the system point of view.

Finally, the literature is plenty of articles concerning the radar-based hand gesture by exploiting neural networks thus the novelty of the present paper should be highlighted compared to the literature.

Reviewer 2 Report

The manuscript reports a Spiking Neural Network approach for the recognition of hand gesture from available Soli and Dop-NET reference datasets. The Authors achieve over 98% accuracy on 10-fold cross-validation for these two datasets, with only 8 machines of less than 1000 neurons. The Authors reports a nice piece of engineering work, which may interest the scientific community involved in applications of machine learning and deep learning approach to real-time processing of radar data. They demonstrate clearly the benefit of the proposed Neural Network approach for radar-based hand gesture recognition. The description by the Authors of this approach is clearly detailed, and the references are sufficient to locate the work with respect to the state-of-the-art in the field of radar-based hand gesture recognition from available Soli and Dop-NET reference datasets. 

Reviewer 3 Report

The manuscript describes a hand gesture recognition system using radar sensors. The proposed method used spiking neural networks and showed improved recognition accuracy with LSM having less than 1000 neurons.
The research and proposed system is interesting, however the experiments are limited to only two datasets. Therefore, it is not clear how feasible the proposed pipeline would perform in a real world setup. The introduction, method and dataset sections are well written, however the reviewer recommends adding some details and providing clarification to the following points:
1. How was the threshold theta value of 0 determined in equation 1?
2. Please add the (LSM) abbreviation next to liquid state machine in the abstract where it is mentioned for the first time.
3. There are more gestures in the SOLI dataset (11) compared to the DOP-NET(4). Was the proposed method used to evaluate only the common gestures between the two datasets?
4. The reviewer recommends adding parameter details about the three classifiers (random forest, SVM and logistic regression). 
For example, what were the slack variable values used in SVM?
5. It is not clear from Fig 1, that LSM is used and in which stage it is used. Please add a label so that the link between
Fig 1 and Fig 6 is more clear.
Overall, the paper is well written and therefore the reviewer recommends an accept after minor revisions.

Round 2

Reviewer 1 Report

The authors have not accomplished my requests.

In detail Figs. 2 and 3 still do not communicate interesting information. The unit is missing both in x and y axis. The first subfigure of Fig. 2 (you should use letters to discriminate the different subfigures) from frame 0 to 4 show the same result, that I cannot interpretate by the way. This because neither the experimental setup, nor the radar processing steps are described. I understand that you are using an existent dataset but you should be aware of how the data were extracted. I have similar concerns regarding Figs 3 and 4.

Author Response

The authors have not accomplished my requests.

Point 1: In detail Figs. 2 and 3 still do not communicate interesting information. The unit is missing both in x and y axis. The first subfigure of Fig. 2 (you should use letters to discriminate the different subfigures) from frame 0 to 4 show the same result, that I cannot interpretate by the way. This because neither the experimental setup, nor the radar processing steps are described. I understand that you are using an existent dataset but you should be aware of how the data were extracted. I have similar concerns regarding Figs 3 and 4.

Response 1:  We have added Section 2.1, which details the specifics of FMCW radar processing from transmitted signal to the derivation of range-Doppler and micro-Doppler maps. 

Moreover:

  • All changes are marked with track change [R1Q1] in orange.
  • We have added recent references on specifics of radar processing and its use in relevant applications 
  • Added the units used on range-Doppler maps.
  • Added letters to refer to the subfigures.

Round 3

Reviewer 1 Report

The authors tried to address my concerns. However the technical soundness of the paper remains dubious. Indeed the obtained results scarcely improve the radar literature and the main findings are focused on the neural networks.